# Evaluation of the New Singularity^TM^ Air versus Ambu^®^ Aura Gain^TM^: A Randomized, Crossover Mannequin Study

**DOI:** 10.3390/jcm11247266

**Published:** 2022-12-07

**Authors:** Lukas Gasteiger, Rouven Hornung, Simon Woyke, Elisabeth Hoerner, Sabrina Neururer, Berthold Moser

**Affiliations:** 1Department of Anaesthesiology and Intensive Care Medicine, Medical University of Innsbruck, 6020 Innsbruck, Austria; 2Department of Clinical Epidemiology, Tyrolean Federal Institute for Integrated Care, Tirol Kliniken GmbH, 6020 Innsbruck, Austria; 3Department of Anaesthesiology and Intensive Care Medicine, See-Spital Horgen, 8810 Horgen, Switzerland

**Keywords:** supraglottic airway device, airway management, difficult airway management, laryngeal mask, mannequin

## Abstract

Background: This randomised crossover mannequin study aimed to compare the insertion time for the newly developed Singularity^TM^ Air and the Ambu^®^ AuraGain^TM^. The Singularity^TM^ Air includes a bendable tube in order to allow optimal passform. Methods: Fifty anaesthetists with a minimum of 100 supraglottic airway device insertions were recruited and randomly assigned to start either with the Singularity^TM^ Air or with the Ambu^®^ AuraGain^TM^. Participants watched a tutorial video the day before the assessment and received a standardized introduction immediately before the assessment. The primary outcome was the time for successful insertion. Secondary parameters were the overall insertion success rate, the numbers of insertion attempts (maximum three), the glottic view through a flexible bronchoscope, and the success rate for gastric tube insertion. Results: Fifty participants were eventually recruited and randomly assigned to insert both devices according to the randomization. The insertion time was 24 s for Singularity^TM^ Air as compared to 20 s for Ambu^®^ AuraGain^TM^ (*p* < 0.001). Overall insertion rate was 92% for the Singularity^TM^ Air as compared to 100% for the Ambu^®^ AuraGain^TM^ (*p* could not be derived as one variable is a constant). The primary insertion success rate was better for the Ambu^®^ AuraGain^TM^ than for the Singularity^TM^ Air (94% versus 68%; *p:* 0.002, respectively). Conclusion: The time for successful insertion and the insertion success rate for the newly developed Singularity^TM^ Air is inferior to that for the Ambu^®^ AuraGain^TM^.

## 1. Introduction

The introduction of the Classic Laryngeal Mask (cLMA) closed an important gap between face mask ventilation and endotracheal intubation [1]. With introduction of the ProSeal^TM^ LMA as the first second-generation supraglottic airway (SGA) device, the use of supraglottic airways became considerably safer, thanks to an oesophageal drain that reduces the risk for gastric insufflation and regurgitation, as well as an improved airway seal that allows much higher airway pressures [2,3,4]. In the meantime, many different types of second-generation SGAs have been put on the market and assessed for their safety and performance in patients undergoing general anaesthesia for elective surgery [5,6,7,8,9].

The possibility of performing bronchoscopically guided intubation has become an important feature in newer SGAs such as the Ambu^®^ AuraGain^TM^ (Ambu^®^, Ballerup, Denmark) or the LMA^®^ Protector^TM^ Airway (Teleflex Medical Europe Ltd., Athlone, County Westmeath, Ireland). In fact, bronchoscopically assisted intubation with an SGA is, nowadays, a technique that is well established and recommended by official bodies for the management of difficult airways [10,11].

Recently, different articles have theorised about how a third-generation SGA could look [12,13]. The authors described a device that should incorporate all the features of a second-generation SGA, and which would have the ‘added facility of correct placement under direct vision’.

The Singularity^TM^ Air (Singularity AG, Maur/Zurich, Switzerland) is a newly developed second-generation SGA for single use and CE-approved for clinical use [14]. The tube’s curvature can be operated from outside the patient, as it has a bending mechanism integrated in the connector part. This allows simple tube flexion and extension by turning an activating wheel. With this feature it should be possible to optimize the position inside the pharynx, thereby increasing oropharyngeal leak pressure and easing the endotracheal tube passage. With the latter, it is promoted by the manufacturer as a possible further evolution of a second-generation SGA [14]. To the best of our knowledge the Singularity^TM^ has not yet been evaluated in any published studies.

The aim of this crossover mannequin study was to assess the success rate and time of insertion for the Singularity^TM^ Air when used by resident and consultant anaesthetists as compared to those for the Ambu^®^ AuraGain^TM^ laryngeal mask. Feasibility for insertion of a gastric tube and quality of flexible bronchoscopic view of the laryngeal inlet were also assessed.

## 2. Materials and Methods

Approval by the local ethics committee was obtained before study initiation (EK Nr: 1225/20, dated 30 December 2020, Ethikkommission der Medizinischen Universität Innsbruck, Austria). This randomised, non-blinded, crossover study was performed in April 2021 at the Department of Anaesthesiology and Intensive Care Medicine at Innsbruck Medical University Hospital, Innsbruck, Austria.

### 2.1. Singularity^TM^ Air

The Singularity^TM^ Air is a further improved CE approved and commercially available single-use second-generation SGA. Up to now, the Singularity^TM^ Air has not yet been tested in any controlled and published trials. The Singularity^TM^ Air offers a separate access to the airway and the digestive tract. The Singularity^TM^ Air is promoted as having a flat design in the itself-bendable tube, which leads directly into the mask with an inflatable cuff in order to adapt to the structures of the hypopharynx, with the opening facing the larynx. A bending mechanism with a turning wheel integrated in the connector allows both flexion and deflexion of the SAG tube (Figure 1). According to the manufacturer, the possibility of improving the position through the bending mechanism should also allow for bronchoscopic guided tracheal intubation. The gastric lumen extends from a color-coded yellow angled connector at the proximal end of the SGA to the tip of the laryngeal mask. The angled connector should therefore allow for comfortable access, even when large airway filters are used. Moreover, according to the manufacturer, the lumen enables insertion of a well-lubricated gastric tube for the evacuation of gastric fluids and gases. According to its description, the customizable bending tube enables easier insertion and positioning of the laryngeal mask, even when changing the position of the patient’s head. The integrated bite protection should reduce potential harm to and obstruction of the tube. The inflatable line ending in a pilot balloon is responsible for in- and deflation of the laryngeal cuff. All parts of the Singularity^TM^ Air are latex-free [14].

### 2.2. Ambu^®^ Aura Gain^TM^

The Ambu^®^ Aura Gain^TM^ (Ambu^®^, Ballerup, Denmark) is a second-generation SGA with integrated gastric access (Figure 2). In several studies, insertion rates and oropharyngeal leak pressures have been assessed and shown to be comparable to those of other second-generation SGAs [8,15]. The mask itself is anatomically curved and it has been shown that the device also provides a safe and effective option for fibreoptic bronchoscopy (FOB)-guided intubation in adults and children [16,17]. The Ambu^®^ Aura Gain^TM^ is used at our institution as a rescue SGA when the routinely used (first-generation) SGAs show insufficient performance.

### 2.3. Participants

All resident and consultant anaesthesiologists from the Department of Anaesthesiology and Intensive Care Medicine, Innsbruck, Austria were invited to participate in this study. Inclusion criteria were minimal professional experience of six months, and more than 100 SGA insertions in patients according to their regular semi-annual activity assessment. Exclusion criteria were refusal to participate and less than six months of anaesthesiologic training.

After that, a video tutorial explaining the insertion technique and the trial were sent to all anaesthesiologists at the Department of Anaesthesiology and Intensive Care Medicine at Innsbruck Medical University Hospital the day before the assessment took place. Fifty anaesthesiologists were ultimately included in the trial after written informed consent was obtained from all participants. Participants were then randomly assigned using the algorithm provided by www.randomization.com (accesses on 14 October 2020). Sealed and numbered opaque envelopes, which contained the random allocation, were opened immediately after that. Each participant received standardized instruction from one of the study authors immediately before assessment. Then, both devices were inserted consecutively from each participant according to randomisation to one of the two groups (Singularity group or Ambu AuraGain group), defining which device was inserted and assessed first.

### 2.4. Study Design

Insertion of both SGAs was performed in a randomised order with regard to the SGA brand and each participant (www.randomization.org, (accesses on 14 October 2020)). Only size #4 SGAs were used. The back of each SGA was lubricated with a water-soluble gel (K-Y^®^, Johnson & Johnson™, Les Moulineaux, France) and the cuff was fully deflated before starting the assessment. The mannequin used for this trial was the Laerdal Airway Management Trainer^TM^ (Laerdal Medical, Stavanger, Norway). Each airway device was inserted according to the manufacturer’s recommendations. Insertion time was defined as the interval between picking up the prepared device, its successful placement and effective ventilation of the mannequin’s lungs, and included up to three attempts. If the insertion time exceeded 120 s, the assessment was stopped. After 30 s, the attempt was determined to be a failure and a second insertion attempt started without stopping the time. For the third and final attempt, 60 s were allowed. The Singularity^TM^ Air was slightly flexed for the first and fully flexed for the second and third insertion attempts in the mannequin. After successful insertion, a set cuff pressure of 60 cmH_2_O was chosen [6,7,18,19]. Effective ventilation was determined by visually checking for any chest wall movement. If adequate ventilation could not be achieved, the insertion was also rated a “failed insertion.” The aetiology of the failure was then noted and designated as one of the following three causes: A.Insertion not possibleB.Insertion possible, ventilation not possibleC.Any other reason.

Once ventilation was achieved, a gastric tube was inserted through the designated lubricated channel. 

Then, a flexible bronchoscope with an outer diameter of 4.8 mm (OlympusTM BF-Q190; Volketswil, Switzerland) was inserted. The fiberscope was advanced to the end of the exit from the laryngeal mask, thereby documenting the Brimacombe score (1 = no vocal cords visible, 2 = vocal cords and anterior epiglottis visible, 3 = vocal cords and posterior epiglottis visible, 4 = vocal cords visible) [20]. For the Singularity^TM^ Air two such scores were recorded, one in flexed and one in relaxed mode.

### 2.5. Outcomes

The primary outcome parameter was the time needed for successful insertion.

Secondary outcome parameters were the overall insertion success rate for the insertion of the SGA, with a maximum of three attempts, the number of insertion attempts needed to insert the SGA, and the success rate for gastric tube insertion.

For the insertion of a gastric tube, the subjective ease of insertion was registered using a four-point Likert scale (1 = easy passage, 2 = passage with minimal resistance, 3 = passage possible with strong resistance, 4 = passage not possible).

Upon successful insertion of the SGA, the above-mentioned Brimacombe score was used to assess visualization of the larynx through the device. This was performed by each participant and supervised by a study author on a screen [20].

### 2.6. Statistics

Statistical analyses were made using IBM SPSS^®^ 27 (IBM Corp. Released 2020. IBM^®^ SPSS Statistics for Windows, Version 27.0. Armonk, NY: IBM Corp). Categorical data are presented as numbers and percentages, while numerical data are given as median (IQR). The data were pre-tested to check the assumption of negligible carryover effects (within-subject sums of the results from both periods were tested) and to check for negligible period effects (within-subject differences of the results from Period 1 and Period 2 were tested). No evidence of relevant carryover effects or period effects was detected. Mann–Whitney U tests were used for pre-tests and effect tests. Secondary analyses were performed using McNemar tests. *p* values < 0.05 were considered statistically significant.

Our a priori sample size calculation, performed in G*Power for Mac, is based on ANOVA using previously published data [21]. A sample size of 50 participants was determined a priori based on a 1:1 treatment allocation. The study has a power of 80% to detect a decrease of 20% in the time to SGA insertion, with a two-sided significance level of 5%. A dropout rate of up to 10% was estimated in our sample size calculation.

## 3. Results

Sixty-seven anaesthetists from our institution responded to our invitation and were assessed for eligibility. One colleague declined to participate and four did not meet the inclusion criteria. Another 10 colleagues, who willing to participate, were absent on the days the trial was realized (Appendix A). A total of 50 participants inserted both SGAs in randomised order according to assigned the group (“Singularity^TM^ first insertion group” or the “Ambu^®^ AuraGain^TM^ first insertion group”) after written informed consent was granted. Baseline characteristics of the participants are summarized in Table 1.

A significant difference in the insertion time was seen between the Ambu^®^ Aura Gain^TM^ and the Singularity^TM^ Air (20.0 s (IQR 17.8–22.0 s) versus 24.0 s (IQR: 21.0–36.0 s); *p* < 0.001; see also Table 2).

The insertion success rate was significantly higher for the Ambu^®^ Aura Gain^TM^ than for the Singularity^TM^ Air (100% vs. 92%, respectively; significance test could not be performed because one variable is a constant).

Moreover, the first-attempt insertion rate was higher for the Ambu^®^ Aura Gain^TM^ than for the Singularity^TM^ Air (94% versus 68%, respectively; *p*: 0.002; see Table 2). No significant difference was found for second or third attempts. All failed insertion attempts using either type of SGA were ascribed to cause A: “Insertion not possible.” All inserted masks allowed sufficient ventilation of the mannequin.

No significant difference was found for ease of insertion of the SGA or for the success rate for gastric tube insertion (Table 2).

Optimal endoscopic view of the larynx (Grade 4: only vocal cords visible) was found in 95.7% of insertions with the Singularity ^TM^ Air and in 98% of those with the Ambu^®^ Aura Gain^TM^ (not significant).

## 4. Discussion

This mannequin-based crossover study evaluated the insertion characteristics of the newly developed Singularity^TM^ Air as compared to those of the Ambu^®^ Aura Gain^TM^ when used by experienced anaesthesiologists. The main finding was that the time for successful insertion and the insertion success rate for the Singularity^TM^ Air was significantly inferior compared to the Ambu^®^ Aura Gain^TM^. The insertion success rate and the insertion time for the Ambu^®^ Aura Gain^TM^ laryngeal mask are comparable to those in reported previous mannequin-based studies [21,22]. To date, there are no comparable data for the Singularity^TM^ Air as this is the first controlled trial.

Over the last decades, the evolution of SGA has happened. Today, so-called second generation SGAs such as the I-Gel^®^, the LMA Supreme™ or the Ambu^®^ AuraGain™ have become widely available. They provide a second tube to insert a gastric tube and also provide a better seal compared to the older models [5,6,7,15]. Another important improvement was the introduction of SGAs specially designed to allow bronchoscopically assisted tracheal intubation, such as the Ambu^®^ Aura –i™, the air-Q^®^ or the Ambu^®^ Aura Gain™ [17,23]. The bronchoscopically assisted tracheal intubation has become a central element in the management of the expected and the unexpected difficult airways in both adults and children [10,24].

The Singularity^TM^ Air has become a further alternative available since its introduction to the market. According to the manufacturer, through its flexion and extension mode it is designed to allow for an optimalization of the position in the hypo-pharynx under bronchoscopic view, and it is promoted by the manufacturer as allowing for a better seal and for bronchoscopic guided tracheal intubation [14].

In this mannequin study, insertion time was significantly longer for the Singularity^TM^ Air. However, a difference of 4 s, even if statistically significant, is clearly not clinically relevant and both insertion times are acceptable. This said, in our trial the Singularity^TM^ Air has shown some important disadvantages as compared to the Ambu^®^ Aura Gain^TM^. A first-attempt insertion rate of 68% is considerably lower than the 94% for the Ambu^®^ Aura Gain^TM^ and is certainly unsatisfactory when compared to other studies [25,26]. A possible explanation for this may be found in the soft tip of the Singularity^TM^ Air. Indeed, the bendable bar that allows flexion and extension ends at the level of the distal orifice in the middle of the cuff. Therefore, at this level there is a sort of predetermined kinking point, which was the main reason for insertion failure. It should also be noted that, in the four failed insertions at this point of the SGA, there was a suspicion of dehiscence in the laryngeal cuff. This might be one possible suggestion for improvement of the newly developed Singularity^TM^ Air, as this could suggest that this is a part where high mechanical stress can act on the SGA and could be a weak point that can influence the performance and the reliability of the Singularity^TM^ Air. Another explanation could be that the mask itself seems a little bulkier than the Ambu^®^ Aura Gain^TM^, and therefor may have a worse fit for the mannequin used.

The fact that measured insertion times for the Singularity^TM^ Air varied between 10 and 93 s among the different users demonstrates that, for some participants, insertion of the Singularity^TM^ Air felt rather easy, indicating a potential learning effect. Interestingly, if the SGA was inserted, effective ventilation was always possible. The flexion and extension modes themselves did not show any effect in this trial, but this may be due to different tissue properties of the mannequin when compared to real patients [27].

The main limitation of this study is that an airway mannequin is not a true anatomical representation of the upper airway, but may have tissue properties (rigidity, compliance) and anatomical dimensions that differ from those of a ‘real’ patient’s oropharyngeal cavity [27]. Therefore, this study only reflects a possible application of the bendable tube in an in-vitro setting. In general, as a mannequin-based study, it does not allow any conclusions on use of the SGAs in patients due to possible differences in tissue properties or anatomy. Therefore, clinical trials in patients are needed to further evaluate the performance of the Singularity^TM^ Air.

## 5. Conclusions

In conclusion, the Singularity^TM^ Air is an interesting new device with a unique feature that distinguishes this mask from all the others. Despite this, our results indicate that further improvement of the new Singularity^TM^ may be needed to provide optimal efficacy.

## Figures and Tables

**Figure 1 jcm-11-07266-f001:**
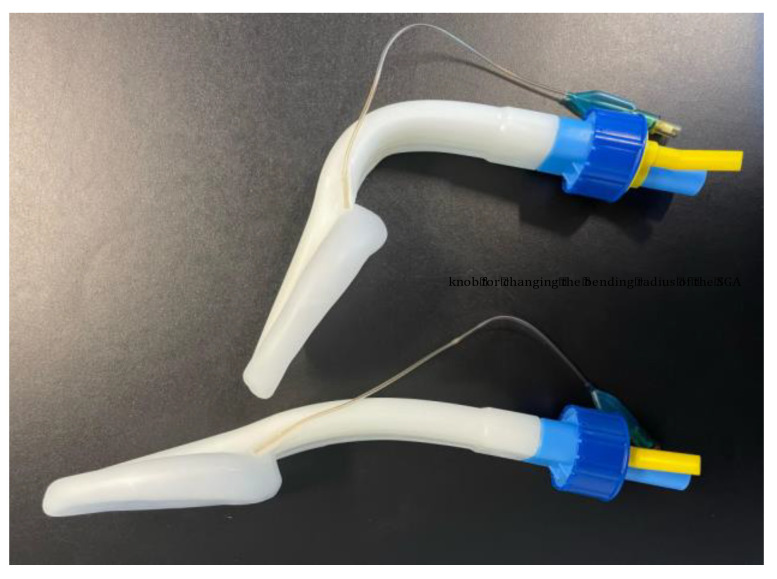
Singularity^TM^ Air Supraglottic Airway Device.

**Figure 2 jcm-11-07266-f002:**
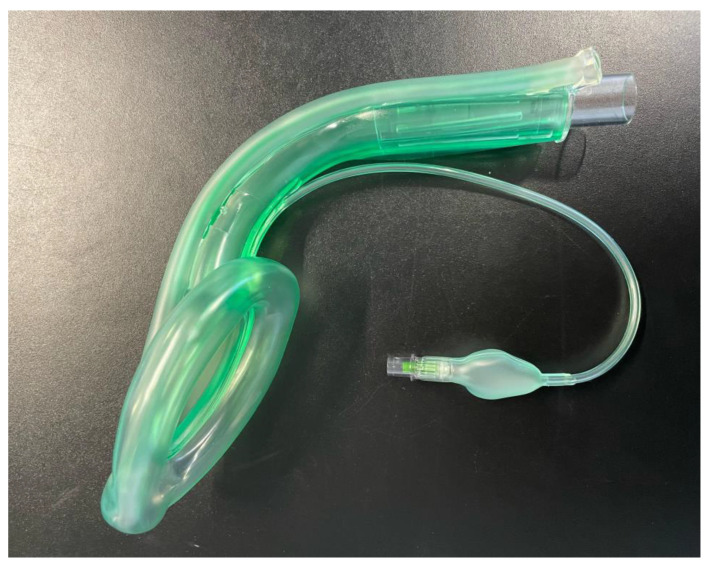
Ambu^®^ Aura Gain^TM^.

**Table 1 jcm-11-07266-t001:** Baseline characteristics of participants.

Number of Participants	50
Gender (f/m): *n* (%)	14 (28/36/72)
Anaesthesiologic experience: *n* (%)	
6–24 months	5 (10)
25–48 months	15 (30)
49–72 months	17 (34)
73–120 months	13 (26)
Watched video tutorial: *n* (%)	50 (100)
Received standardized introduction: *n* (%)	50 (100)

**Table 2 jcm-11-07266-t002:** Insertion success, insertion time, fibreoptic view to the glottic, aetiology of failed insertion, ease of insertion, insertion success rate for gastric tube, aetiology of failed insertion gastric tube. Data are mean or numbers (%).

*n*	Singularity^TM^ Air	Ambu^®^ AuraGain^TM^	*p*
PRIMARY VARIABLE			
Insertion time; seconds	24.0 (IQR: 21.0–36.0) [10–93]	20.0 (IQR: 17.8–22.0) [10–61]	<0.001
SECONDARY VARIABLES			
Overall insertion success rate	46 (92%)	50 (100%)	-^†^)
First attempt	34 (68%)	47 (94%)	0.002
Second attempt	7 (14%)	2 (4%)	0.921
Third attempt	5 (10%)	1 (2%)	1.000
Fail	4 (8%)	0 (0%)	-
Ventilation			
Fibreoptics position airway tube * 4/3/2/1; *n* (%)			
		49 (98)/1(2)/0/0	
Flexed mode	44 (95.7)/2 (4.3)/0/0		1.000
Relaxed mode	44 (95.7)/2 (4.3)/0/0		1.000
Aetiology of failure ^ A/B/C; *n*	11/0/0	2/0/0	

^†^ Mc Nemartest could not be performed because one variable is a constant; * 4 = only vocal chords visible; 3 = vocal chords plus posterior epiglottis; 2 = vocal chors plus anterior epiglottis; 1 = vocal chords not seen; ^^^ A: insertion not possible; B: SAG insereted, ventilation not possible; C: any other reason.

## Data Availability

The datasets analysed during the current study and the statistical code will be available from the corresponding author on reasonable request and will only be accessible to personnel directly involved in the study.

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
