# Peer review of "Evaluation of the New SingularityTM Air versus Ambu® Aura GainTM: A Randomized, Crossover Mannequin Study"

_jcm, 2022, doi:10.3390/jcm11247266_

Round 1
Reviewer 1 Report
the article is well written, clear, concise. the tables and pictures are appropriate. the conclusions are supported by the data.
50 anesthetists were included in the study. each of them inserted just one of each laryngeal masks. can the authors comment on why they chose this design? one would expect that each anesthetist could inserte e.g. 10 laryngeal masks. usually after several attemps with one device one becomes more familiar with the handling. can the authors also comment on the primary outcome, time to insertion. this was 20 s 24 sec, which is statistically significant. but a difference of 4 sec is completely irrelevant in clinical practice. it do not think this is a factor that merits consideration in clinical practice.
Author Response
Reviewer #1:
- Comment 1:
the article is well written, clear, concise. the tables and pictures are appropriate. the conclusions are supported by the data.
50 anesthetists were included in the study. each of them inserted just one of each laryngeal masks. can the authors comment on why they chose this design? one would expect that each anesthetist could insert e.g. 10 laryngeal masks. usually after several attempts with one device one becomes more familiar with the handling.
Response:
We apologize we did not describe this clearer. All fifty participants inserted both devices in a cross-over study according to randomization. The randomization only defines the device (Singularity TM or Ambu AuraGain) that was inserted first. We choose this cross-over design in order to eliminate the bias you mention that you may perform better with the second device. We also considered this “learning effect” in our statistics in order to eliminate any possible carry-over effect.
We now specify accordingly in the Abstract section as follows:
“…. Fifty anaesthesiologists with a minimum of 100 supraglottic airway device insertions were recruited and randomly assigned to start either with the SingularityTM Air or with the Ambu® AuraGainTM. “(Line 12-14)
Also:
“Fifty participants were eventually recruited and randomly assigned to insert both devices according to the randomization.” (Line 19 -20)
And in the Methods section as follows:
“Participants were then randomly assigned using the algorithm provided by www.randomization.com. Sealed and numbered opaque envelopes, that contained the random allocation, were opened immediately after that each participant received standardized instruction from one of the study authors immediately before assessment. Then both devices were inserted consecutively from each participant according to randomization to one of the two groups (Singularity group or Ambu AuraGain group) defining which device was inserted and assessed first.” (Line 119-125)
Also in the Methods section as follows:
“The data were pre-tested to check the assumption of negligible carryover effects (within-subject sums of the results from both periods were tested) and to check for negligible period effects (within-subject differences of the results from Period 1 and Period 2 were tested). No evidence of relevant carryover effects or period effects was detected.” (Line 178 – 192)
- Comment 2:
Can the authors also comment on the primary outcome, time to insertion. this was 20 s 24 sec, which is statistically significant. but a difference of 4 sec is completely irrelevant in clinical practice. it do not think this is a factor that merits consideration in clinical practice.
Response:
Thank you for this good comment. In fact, a difference of 4 seconds does not make any difference in clinical practice. The major points can be found in our secondary outcome, namely a first insertion rate of just 68% that is clearly not satisfying for a new SGA when inserted by anaesthetists.
We now specified accordingly:
“In this mannequin study insertion time was significantly longer for the SingularityTM Air. However, a difference of 4 seconds even if statistically significant is clearly clinical not relevant and both insertion times are acceptable. This said, in our trial the SingularityTM Air has shown some important disadvantages as compared to the Ambu® Aura GainTM. A first-attempt insertion rate of 68% is considerably lower than the 94% for the Ambu® Aura GainTM and is certainly unsatisfactory when compared to other studies.[25, 26]” (Line 243 – 282)

Reviewer 2 Report
The proposed paper addresses an interesting topic concerning the development of novel supraglottic airway devices.
However, I find several issues with the methodology and description of the results.
1. The introduction refers to the Singularity Air device, describing it and its functions. No references are provided on previous studies validating the device for clinical use, thus making the feasibility of insertion as compared to currently used devices relevant. These references should be provided.
2. Row 54 – it is stated that the device promotes itself as a possible example of a third generation SGA – how?, where?
3. The materials and methods section has several problems:
a. Again, no references are provided on the prior validation of the Singularity Air device or the provenance of the information on the device
b. The recruitment of the study participants is inadequately described – it is not described how many potential participants were excluded from the study and on what basis; furthermore, it is stated that only physicians with more than 100 SGA insertions were included (how was this assessed, is there a registry for each physician?)
c. The title for section 2.4 is rather inappropriate, as it describes the design of the experiment, more than data collection itself
d. The references in row 121 are inadequately placed in this section, as they are completely not related to the sentences they follow. It is not customary to provide references in the material and methods section.
e. There is an important flaw in the description of the experiment design, as there is no information provided on which type of mannequin the study was performed. This is not irrelevant, as they are not interchangeable or identical anatomically.
4. In the statistical analysis section, there is a paragraph discussing period effects, where the authors discuss within-subject differences from Period 1 and Period 2, although having never stated the study was performed in two distinct periods, why that might have been so and what those periods were.
5. The discussion section should address on a point by point basis the results of the experiment, rather than do a review of the published literature, re-stating many of the affirmations made in the introduction.
6. The main 2 limitation of this study are that it compares an un-tested device on a mannequin.
The fact that the device is un-tested make the relevance of the experiment for the public very limited, especially in terms of ease of insertion, as it is unclear whether any interested parties will be faced with the possibility of ever using it.
The second limitation, the testing on the mannequin, is also important. There is data that mannequins are inadequate for testing and developing airway devices, which further limits the applicability and relevance of the results.
Author Response
Reviewer #2
The proposed paper addresses an interesting topic concerning the development of novel supraglottic airway devices.
However, I find several issues with the methodology and description of the results.
- Comment 1:
The introduction refers to the Singularity Air device, describing it and its functions. No references are provided on previous studies validating the device for clinical use, thus making the feasibility of insertion as compared to currently used devices relevant. These references should be provided.
Response:
Thank you for this annotation. Indeed, by now to the best of our knowledge this is the first controlled study assessing the SingularityTM. The product was introduced some years ago, underwent CE approval and is commercially available for clinical use. By now it was not evaluated in any controlled study. Therefore we obviously cannot refer to this. However we now refer to the manufacturers homepage that describes the product.
We specified accordingly:
“The SingularityTM Air (Singularity AG, Maur/Zurich | Switzerland) is a newly developed second-generation SGA for single use and CE-approved for clinical use.[14]” (Line 54 – 55)
And:
“To the best of our the SingularityTM has not yet been evaluated in any published studies.”
(Line: 61 -62)
And:
“The SingularityTM Air is a further improved CE approved and commercially available, single-use second-generation SGA.” (Line 75 – 76)
- Comment 2:
Row 54 – it is stated that the device promotes itself as a possible example of a third generation SGA – how?, where?
Response:
This is a good comment. In fact the term “third Generation SGA” is not yet defined. However recently the discussion stared what a third generation SGA could be like. However, we hampered our statement by changing it to:
“a possible further evolution of a second-generation SGA.” (Line: 60 - 61)
- Comment 3:
The materials and methods section has several problems:
- Again, no references are provided on the prior validation of the Singularity Air device or the provenance of the information on the device.
Response:
As explained in the response to Comment 1 this is the first controlled study assessing the performance of this new device. The information on the device itself are from the company’s homepage. We now added this as a reference. (Line: 55, 61, 91)
- The recruitment of the study participants is inadequately described – it is not described how many potential participants were excluded from the study and on what basis; furthermore, it is stated that only physicians with more than 100 SGA insertions were included (how was this assessed, is there a registry for each physician?
Response:
Thank you for this annotation. All anaesthetists of our institution were invited (109 consultant + 84 residents) according to our website https://anaesthesie.tirol-kliniken.at/page.cfm?vpath=team; date 12_11_2022).
The inclusion criteria were a minimum experience of at least 6 months and as you mentioned more than 100 SGA insertions. The participants were asked to provide this data according to their in-house evaluation, that assess the activity of each anaesthetist based on our PDMS data semi-annually. Of those who responded to our invitation (n = 76), only 4 were excluded because they did not match with our inclusion criteria (< 6 months). One colleague refused to participate and other 10 who responded were absent during the assessment period. The details are now shown in a Consort Flow Diagram. We stopped recruiting when we reached 50 participants as previously determined in our sample size calculation.
We now specified as follows:
” Inclusion criteria were minimal professional experience of at least six months and more than 100 SGA insertions in patients according to their regular semi-annual activity assessment.” (Line 112 - 113)
And:
“Sixty-seven anaesthetists of our institution responded to our invitation and were assessed for eligibility. One colleague declined to participate and 4 did not meet the inclusion criteria. Other 10 colleagues willing to participate were absent on the days the trial was realized (Supplemental Table 1). “ (Line 191 – 193)
- The title for section 2.4 is rather inappropriate, as it describes the design of the experiment, more than data collection itself
Response:
Thank you for this annotation. In fact, this section describes the study design we now changed the subheading accordingly.
“2.4 Study design”
- The references in row 121 are inadequately placed in this section, as they are completely not related to the sentences they follow. It is not customary to provide references in the material and methods section.
Response:
In fact, there are two references that do not refer to the sentence. The other two underline the feasibility to perform a bronchoscopic guided tracheal intubation through the Ambu AuraGain and refer to the sentence they follow. As you mentioned in your comment 1 and comment 3a it should be stated were information that are provided to describe the material used should be provided. Therefore, we decided that the remaining two references are helpful for the understanding of the article.
“[16, 17]” (Line 103)
- There is an important flaw in the description of the experiment design, as there is no information provided on which type of mannequin the study was performed. This is not irrelevant, as they are not interchangeable or identical anatomically.
Response:
We apologize this should not happen. We now added the type of mannequin (Laerdal Airway management trainer).
“The mannequin used for this trial was the Laerdal Airway Management TrainerTM (Laerdal Medical, Stavanger, Norway).” (Line 131 - 132)
- Comment 4:
In the statistical analysis section, there is a paragraph discussing period effects, where the authors discuss within-subject differences from Period 1 and Period 2, although having never stated the study was performed in two distinct periods, why that might have been so and what those periods were.
Response:
This is in fact misleading as we obviously did not describe clearly enough that every participant introduced both SGA consecutively according to the randomization. This was also mentioned by another reviewer. We now corrected and better specified in the abstract and the methods section. We now specify accordingly in the Abstract section as follows:
“…. Fifty anaesthesiologists with a minimum of 100 supraglottic airway device insertions were recruited and randomly assigned to start with either the SingularityTM Air or the Ambu® AuraGainTM. “(Line 12-14)
And in the Methods section as follows:
“Participants were then randomly assigned using the algorithm provided by www.randomization.com. Sealed and numbered opaque envelopes, that contained the random allocation, were opened immediately after that each participant received standardized instruction from one of the study authors immediately before assessment. Then both devices were inserted consecutively from each participant according to randomization to one of the two groups (Singularity group or Ambu AuraGain group) defining which device was inserted and assessed first.” (Line 119- 125)
The mentioned carry over effect and period effect between the two periods therefor refers to the introduction of the first (period 1) and the second device (period 2). We hope that this explaines the two periods that were mentioned.
- Comment 5:
The discussion section should address on a point by point basis the results of the experiment, rather than do a review of the published literature, re-stating many of the affirmations made in the introduction.
Response:
We have discussed more in detail some of the findings in special the very low first insertion rate of the Singularity Air and left away some of the “review of published literature” that in fact was repetitive after the Introduction. Please find the details in the Discussion section (Line 220 -308)
- Comment 6:
The main 2 limitation of this study are that it compares an un-tested device on a mannequin.
The fact that the device is un-tested make the relevance of the experiment for the public very limited, especially in terms of ease of insertion, as it is unclear whether any interested parties will be faced with the possibility of ever using it.
Response:
Here is some misunderstanding. The SingularityTM Air is a CE certified SGA that since 2020 is available and is actually in clinical use in some hospitals in Switzerland (according to the manufacturer). It has undergone testing for CE certification. However this is the first study evaluating its performance and we decided to start with a mannequin based trial. Actually we are discussing whether or not to design a clinical trial. Therefor we think that this article is if interest to the readers. We have now tried to specify this by referencing to the manufactures homepage and by specifying that as follows:
“The SingularityTM Air is a further improved CE approved and commercially available, single-use second-generation SGA.” (Line 75 -76)
And:
“The SingularityTM Air is a further alternative available since its introduction to market. Through its flexion and extension mode it is designed to allow for an optimalization of the position in the hypo-pharynx under bronchoscopic view and is promoted by the manufacturer to allow for a better seal and for bronchoscopic guided tracheal intubation.[14]“ (Line 238 – 242)
- Comment 7:
The second limitation, the testing on the mannequin, is also important. There is data that mannequins are inadequate for testing and developing airway devices, which further limits the applicability and relevance of the results.
Response:
This in fact an important limitation that we have described as such in the discussion section as follows.
“The main limitation of this study is that an airway mannequin is not a true anatomical effigy of the upper airway, but may have tissue properties (rigidity, compliance) and anatomical dimensions that differ from those of a 'real' patient’s oropharyngeal cavity.[27] Therefore this study only reflects a possible application of the bendable tube in an in-vitro setting. In general, as a mannequin-based study, it does not allow any conclusion on use of the SGAs in patients due to possible differences in tissue properties and anatomy” (Line 302 - 308)

Round 2
Reviewer 2 Report
I think the authors have greatly improved their manuscript.
However, I maintain my opinion that based on the existing literature on the relevance of mannequin studies in terms of evaluation of airway devices, the paper is of limited interest to the readers and brings little information on the actual usefulness of the device in real life.
Author Response
Reviewer #2
- Comment 1:
I think the authors have greatly improved their manuscript.
However, I maintain my opinion that based on the existing literature on the relevance of mannequin studies in terms of evaluation of airway devices, the paper is of limited interest to the readers and brings little information on the actual usefulness of the device in real life.
Response:
Thank you for the compliment. You are absolutely right in your opinion that these results may not reflect the performance of the SingularityTM Air in clinical practice, but they are a starting point in order to evaluate this new device in the attendance for further clinical trials. We further strengthened this limitation in the Discussion as follows:
“Therefore, clinical trials in patients are needed to further evaluate the performance of the SingularityTM Air.” (Page 7, line 278-279)
